# Films of Poly(Hydroxybutyrate) (PHB) and Copper with Antibacterial Activity

**DOI:** 10.3390/polym15132907

**Published:** 2023-06-30

**Authors:** Mayte M. Quispe, María E. Villanueva, Guillermo J. Copello, Olivia V. López, Marcelo A. Villar

**Affiliations:** 1Planta Piloto de Ingeniería Química, PLAPIQUI (UNS-CONICET), Camino La Carrindanga Km 7, Bahía Blanca 8000, Argentina; mquispe@plapiqui.edu.ar (M.M.Q.); olivialopez@plapiqui.edu.ar (O.V.L.); 2Departamento de Ciencias Básicas, Universidad de Luján, Luján 6700, Argentina; emiliavillanueva1@gmail.com; 3Instituto de Química y Metabolismo del Fármaco, IQUIMEFA (UBA-CONICET), Universidad de Buenos Aires, Ciudad Autónoma de Buenos Aires 1113, Argentina; gcopello@ffyb.uba.ar; 4Departamento de Química, Universidad Nacional del Sur, Av. Alem 1253, Bahía Blanca 8000, Argentina; 5Departamento de Ingeniería Química, Universidad Nacional del Sur, Av. Alem 1253, Bahía Blanca 8000, Argentina

**Keywords:** poly(hydroxybutyrate), copper (II) sulfate, antibacterial biomaterials, biomaterials, glycerol tributyrate

## Abstract

Poly(3-hydroxybutyrate), PHB, is a hydrophobic biopolymer with good mechanical and barrier properties. However, neat PHB is a semicrystalline polymer with a relative high degree of crystallinity and poor film properties. In this work, this biopolymer was plasticized with glycerol tributyrate and functionalized with copper (II) sulfate, allowing us to obtain biodegradable antimicrobial flexible films. Films with the minimum inhibitory concentration (MIC) of copper (II) sulfate presented a higher roughness than neat PHB films. The presence of plasticizer significantly improved the copper sulfate diffusion process, which was evidenced by a greater inhibition halo for plasticized materials compared to unplasticized ones, at the same salt concentration. Plasticized PHB with 2.5% copper (II) sulfate inhibited both Gram-positive (*Staphylococcus aureus*) and Gram-negative (*Pseudomona aeruginosa*) bacteria, as determined by the bacterial inhibition halo. In addition, neat PHB films and PHB containing copper (II) sulfate did not show in vitro cytotoxicity in the L-929 cell line. Thus, plasticized PHB functionalized with copper (II) sulfate can be used as biodegradable antimicrobial flexible films for different applications.

## 1. Introduction

Nowadays, different situations such as the COVID-19 global pandemic have led to the need and opportunity to search for new materials with antimicrobial properties by incorporating organic or inorganic active compounds [1]. These materials are attractive for different biomedical applications such as drug release, wound healing dressings, and tissue regeneration, among others [2,3,4,5].

Biopolymer-based dressings are made mainly of polysaccharides, proteins, and lipids, and they are biodegradable, nontoxic, and biocompatible [6]. Polysaccharides are the most employed biopolymers for these systems due to their drug-release and permeability characteristics, as well as their mechanical, chemical, and biological properties [7]. Therefore, they have a high potential for the development of biomedical devices [8,9]. Bactericidal dressings are now available and have the effect of controlling or treating infected wounds. Although their final action is not to generate healing, they are used as a first step to debride (clean) the wound bed and reduce the bacterial colonization or infection of the wound [10].

Innovations in antibacterial technology have led to the development of many modern advanced wound dressings that incorporate antibacterial agents, such as silver, iodine, poly(hexamethylene biguanide) (PHMB), gentian violet, and methylene blue [11,12,13]. Nevertheless, no single antibacterial agent is superior; each plays an important role in the antibacterial toolbox for managing an increased bacterial load in wounds [14]. For this reason, it is important to be able to incorporate diverse antimicrobials into materials for wound healing [15]. Thus, to reduce microbial load, antimicrobial agents are widely used in wound healing, improving the conditions of complex wounds [16]. Among antibacterial components, fungicide agents such as copper sulfate salts can be mentioned. In particular, the biocidal activity of copper ions is recognized in the aqueous medium, mainly the cupric ion (Cu^2+^). Copper biocidal activity is a function of its concentration; it can inhibit bacterial growth at concentrations between 25 and 150 μM or act as bactericidal at higher concentrations [17]. Cupric ions inhibit protein synthesis, alter the cell membrane, induce lipid damage, which is key for the exchange of molecules in the intra- and extracellular environments, and alter the nucleic acids of bacteria and viruses [18,19].

The great variety of wound dressings is due to the specific requirements of different types of wounds. Current strategies in the field of functional dressing design focus on accelerating wound repair, employing materials based on biopolymers. Due to the characteristics of the different types of wounds and the different stages of healing, there is not a single dressing that can be applied efficiently in all situations. Therefore, it is possible to develop and optimize different biomaterials for manufacturing dressings that are biocompatible, in terms of their chemical and physical properties, to satisfy the majority of needs for a particular stage of wound healing [20]. The most complex wounds are the chronic ones that, in general, are treated with dressings made up of two sections: (i) a water-impermeable film permeable to gases that protect the wound from the outside (secondary dressing) and (ii) an internal section in contact with the wound (primary dressing) based on superabsorbent materials, since most of the wounds present a high level of exudation [21]. Although there are many developments of materials based on biopolymers, there is currently no other alternative to the most widely used secondary dressing, polyurethane (PU) foams or films. These dressings are elastic, transparent, and adhesive; they are well-known for their superior strength and favorable biocompatibility [9]. PU films are most used on dry or low exudate and surgical wounds since they maintain a humid environment around the wound, provide thermal insulation, and are convenient to wear [16,21].

Poly(hydroxyalkanoate)s (PHAs) are natural biodegradable polyesters that are synthesized by microorganisms, and they have been shown to be useful in tissue engineering, particularly in applications such as vascular grafts, heart valves, and pericardial patches [22]. Another application is as scaffolds for the regeneration of cardiac tissues [23]. They have also been proven to be effective for applications in the controlled release of drugs, allowing the encapsulation of bioactive compounds such as antibiotics or therapeutic drugs [24]. Poly(3-hydroxybutyrate) (PHB) is probably the most common type of PHAs and the inherent characteristics of this biopolyester and the properties of PHB-based films [25,26] make it interesting as secondary dressings for wound healing, although this application has not yet been investigated in depth.

The aim of this work was to obtain active films based on poly(3-hydroxybutyrate) (PHB), plasticized with glycerol tributyrate (GTB), and copper (II) sulfate as antimicrobial. Materials were characterized in order to study the effect of copper (II) sulfate on film structure, mechanical properties, adhesiveness, cytotoxicity, and antimicrobial capacity.

## 2. Materials and Methods

### 2.1. Materials and Films Obtaining

Poly(3-hydroxybutyrate) powder (PHB) was purchased from Biomer^®^ (Schwalbach, Germany). The additives used were glycerol tributyrate (GTB, 98.5% purity, Sigma Aldrich, Switzerland) as plasticizer, Irganox^®^ 1010 (I, 98% purity, Sigma Aldrich, Saint Louis, MO, USA) as phenolic antioxidant, and copper (II) sulfate pentahydrate CAS [7758-99-8] (Cu, 98% purity, Biopack, Buenos Aires, Argentina) as antimicrobial agent.

The studied formulations are described in Table 1. Samples were first melt-processed in a Brabender Plastograph (Brabender, Duisburg, Germany) at 180 °C and 60 rpm, for 15 min. Films were obtained from processed samples by thermocompression using a hydraulic press at 5 kg cm^−2^ and 195 °C, for 15 min. Thickness was measured using an electronic digital caliper MAX-CAL (Fowler & NSK, Canton, MA, USA).

### 2.2. Films’ Microstructure

Samples were frozen with liquid nitrogen and cryofractured. Then, they were mounted on aluminum stubs exposing the cross section and metalized using an Ar plasma metallizer. The morphology of the films’ cross sections was studied by scanning electron microscopy (SEM), using a LEO EVE 40 XVP (Thornwood, NY, USA) microscope with a secondary electron detector. Moreover, using an Oxford X-max 50 energy-dispersive X-ray microanalysis system (EDS) attached to the microscope, a microanalysis was carried out to get information about the elemental chemical composition of a small area of each sample.

Microphotographs of the surface of the PHB films were obtained by atomic force microscopy (AFM) using a Carl Zeiss focal laser microscope (Oberkochen, Germany). A film specimen (2 × 2 mm) was fixed on a stub by some double-sided adhesive tape. Scanning areas from 10 × 10 μm^2^ to 500 × 500 μm^2^ allowed us to determine the medium surface roughness (Ra). Topographic measurements were carried out using ConfoMap Premium v7 software.

### 2.3. Films’ Tensile Properties

Tensile tests were carried out in an Instron tensile testing machine model 3369 (UK), using a crosshead speed of 5 mm/min and a load cell of 1 kN. The maximum tensile strength (σ_m_), Young’s modulus (E), and elongation at break (ε_B_) were calculated, according to ASTM D882-91.

### 2.4. Films’ Antimicrobial Capacity

Agar diffusion assays were performed on Luria–Bertani (LB) agar to evaluate the antibacterial activity against *Pseudomona aeruginosa* and *Staphylococcus aureus* as model pathogens of Gram-negative and Gram-positive bacteria, respectively. *Staphylococcus aureus* ATCC 29,213 was generously provided by the Microbial Culture Collection of Facultad de Farmacia y Bioquímica (CCM 29), University of Buenos Aires, and wild-type *Pseudomona aeruginosa* was isolated from a hospital environment. All microorganisms were grown at 35 °C for 24 h on Luria−Bertani (LB) medium (Britania, Buenos Aires, Argentina).

Colonies of the previously mentioned bacteria, obtained from an overnight culture, were suspended in LB broth and the concentration was adjusted to 10^5^ CFU mL^−1^. Then, 0.200 mL of this suspension was spread on LB agar plates. Film samples (0.5 cm diameter disks) were disinfected with 1 mL of 70% ethanol for 15 min and used for antibacterial efficacy assays after washing three times with sterilized water. Samples were then rested on the inoculated medium and after incubation at 37 °C for 24 h, the inhibition zone was observed and measured. These assays were performed in triplicate.

### 2.5. Films’ Cytotoxicity

The cytotoxicity of the film components (biopolymer, plasticizer, antioxidant, and antimicrobial agent) was studied in cell cultures. The cell line used was L-929 (mouse fibroblasts) according to ISO 10993-5:2009. L929 cells were from the American Type Culture Collection, from the USA (CCL-1), were grown in D-MEM (Dulbecco’s Modified Eagle Medium, Gibco) supplemented with 5% fetal bovine serum (FBS). Cell viability was evaluated through the cell metabolic activity by reduction of the 3-(4,5-dimethylthiazol-2-yl)-2,5-diphenyltetrazol bromide (MTT). The MTT microtiter quantitative colorimetric method is useful for evaluating viability and survival in cytotoxicity treatments [27]. The tetrazolium reduction assay was the first homogeneous cell viability assay developed for a 96-well format that is suitable for high-throughput screening [28]. The MTT assay was performed in 96 well-plates and L929 cells, passage 89, were seeded at a density of 0.01 × 10^6^ cells/well. The MTT substrate was prepared in a physiologically balanced solution (5 mg/mL), added to the cells in culture, and incubated for 1 to 4 h. The amount of formazan (presumably directly proportional to the number of viable cells) was measured by recording changes in absorbance using a plate reading spectrophotometer DT880 Multimode detector (Beckman Coulter, Brea, CA, USA). The activity of mitochondrial enzymes was related to MTT, resulting as a measurement of mitochondrial activity [28,29].

A sample disk, punched out, was placed in each well. Monolayers were incubated, together with the materials in individual wells, at 37 °C with 5% CO_2_ for 24 h (Dulbecco’s Modified Eagle Medium, Gibco). The CO_2_ used was high-purity anhydride carboxide (Air Liquid, Buenos Aires, Argentina). After this time, materials were extracted and cell cultures were observed under an inverted optical microscope to determine possible morphological alterations in cells that were in contact with the material, compared to the control wells (containing a cell culture medium without the sample).

Before observing under the microscope, 20 μL of MTT was incorporated and after a 3 h incubation at 37 °C, samples were centrifuged at 1500 rpm for 5 min; the supernatant was extracted and 100 μL of a solution of 1/10 formalin/isopropyl alcohol was added to stop the MTT reduction reaction. Subsequently, the optical densities of the films’ monolayers and of negative controls were measured at 450 nm. The comparison of the optical density values for the different concentrations of the product components were carried out by means of an ANOVA and Tukey’s test. These differences were considered significant when *p* ≤ 0.05.

### 2.6. Films’ Adhesive Strength to Human Synthetic Skin

Film adhesive strength to skin was evaluated using a procedure similar to the bioadhesion test of patches reported by Wong et al. [30]. A texture analyzer (TA Instrument, New Castle, DE, USA) equipped with a 50 N load cell was employed to determine the bioadhesion force using human synthetic skin as a model tissue. The human synthetic skin used was a high-performance silicone rubber (Casiopea Beauty Store, Buenos Aires, Argentina), with a thickness of 1000 μm. This material is generally employed to practice tattooing, to create skin effects on medical prosthetics, and for cushioning applications [31]. Circular specimens (2 cm diameter) of the synthetic skin were mounted onto a cylindrical poly(lactic acid) (PLA) support of 2 cm diameter and 4 cm length and secured with double-sided tape. A foam was placed underneath the human synthetic skin on the PLA to provide a cushioning effect. Double-sided tape was placed both between the PLA support and the foam and between the foam and the skin in order to prevent the slipping of materials during testing. The human synthetic skin was further secured by placing a PLA cap over this support. This cap had a circular hole of 17 mm diameter on its top to expose the human synthetic skin to contact with the adhesive during measurements. The whole PLA support was then positioned at the bottom of the measuring system and held in place by screws to the equipment. PHB films were impregnated with a thin layer of a commercial adhesive for skin and hairpieces MASTIC (PINTAFAN, Buenos Aires, Argentina). Then, a film was attached to a PLA support with a similar diameter (17 mm) to the cap hole using double-sided tape, leaving the adhesive layer exposed. Finally, this support was screwed to the upper probe of the instrument. These two PLA supports were aligned to ensure that PHB films and adhesive would come into direct contact with the exposed surface of the human synthetic skin when the upper PHB film support was lowered (Figure 1). All measurements were carried out at room temperature (21 °C) and a relative humidity of 30 to 50%.

The evaluation of bioadhesion performance was conducted using different contact forces (1, 2, and 5 N), a contact time of 300 s (Figure 1a), and a probe speed of 1 mm/s (Figure 1b). For each film, ten replicates were conducted per test. The parameter “peak detachment force” was used to study the adhesiveness of the film and the measured value corresponded to the maximum force required to detach the film from the synthetic skin.

### 2.7. Statistical Analysis

A completely randomized experimental design was used to characterize composite films, using an analysis of variance (ANOVA) to compare the mean differences in samples’ properties by comparing the mean values with Fisher’s least significant difference test with a significance level at *p* = 0.05.

## 3. Results

### 3.1. Films’ Microstructure

To corroborate the presence of copper sulfate particles within the PHB matrix, SEM and EDS were carried out. The corresponding micrographs and spectra are shown in Figure 1a,b, respectively. Copper (II) sulfate crystals were well distinguished from the polymer matrix in all studied samples. It was evidenced that salt particles presented sizes ranging between 25 μm and 50 μm. The elemental analysis of copper (II) sulfate particles showed peaks associated to elemental copper (Cu), in addition to small traces of oxygen (O), suggesting a partial oxidation of particles upon contact with the environment. In addition, the presence of sulfur (S) associated with sulfate ions was also observed.

Roughness is an important property of polymer films, especially for certain applications or surface functions. Figure 2 presents the 2D atomic force micrographs of plasticized PHB films without and with 2.5% copper sulfate, respectively. The incorporation of copper (II) sulfate generated a notable increase in film roughness (Ra), being 45 ± 8 nm and 105 ± 14 nm for PHB-I-0Cu_GTB_ and PHB-I-2.5Cu_GTB_, respectively. This increment could be attributed to the presence of salt microcrystals in the film surface. However, both values were lower than those reported by Chan et al. [32]. A similar study showed that the incorporation of mineral fillers notably increased the average roughness of PHB-based materials [33]. These roughness values for the materials were also comparable to the roughness values reported for commercial deposits, with excellent softness in contact with the skin surface [34].

### 3.2. Films’ Tensile Properties

Filler incorporation within polymer matrices modifies their mechanical properties, increasing the rigidity and resistance of the materials. The final properties of composite materials depend on several factors: particle size, filler concentration, and interfacial particle–matrix adhesion [34]. Figure 3a–c shows the changes in mechanical properties of PHB films by the addition of different concentrations of copper (II) sulfate particles. An increase in the salt concentration led to a decrease in the film’s mechanical strength and elongation at break. These changes could be mainly due to the large dimensions of the added particles (25 to 50 μm). As reported by other authors, particles that are usually used as reinforcements have nanometric dimensions, since no stress force concentration is developed in the polymer matrices, thus improving the material’s mechanical properties [34,35,36]. On the other hand, a decrease in mechanical resistance and elongation at break with an increase in the size of added particles has also been reported [37]. In general, the maximum tensile strength increases when the surface area of the particles decreases, which is the case of nanoparticles giving an efficient stress distribution in the polymer matrix [38]. The obtained results suggest that to improve the mechanical properties of PHB films by adding copper (II) sulfate, it would be necessary to carry out a reduction of solid-size particles prior to the thermal processing. Therefore, grinding salt and a subsequent sieving to obtain a material with a smaller size could be a possible solution.

It is expected that the Young’s modulus of PHB films increase if hard micrometric particles are added to the matrix, improving the material’s stiffness. However, Figure 3c shows that there were no major changes in the values of Young’s modulus with the addition of copper (II) sulfate particles up to 7.5%. For both sets of films, in the range of salt concentrations studied, only a slight decrease was observed at high salt concentrations, which agreed with results reported in the literature using a similar size of mineral particles (20–50 μm) [38]. The particles’ number and size have little effect on material stiffness, as well as the interfacial adhesion of mineral fillers/polymeric matrix when particles of micrometric size and irregular shape are used [35].

### 3.3. Films’ Antimicrobial Capacity

Figure 4 shows the inhibition halo of the PHB films for *S. aureus* and *P. aeruginosa*. A film’s capacity to inhibit microbial development is associated with the diffusion and release of the active substance from the matrix to the medium. Therefore, polymer chains must move cooperatively and thus allow the transport of antimicrobial agents. As it was expected, the PHB films without the copper (II) sulfate did not present any antimicrobial capacity against any of the tested bacteria. All samples containing copper (II) sulfate had the capacity to inhibit the development of *S. aureus* (Figure 4a), regardless of the presence of the plasticizer. As expected, the higher the concentration of the antimicrobial agent the greater the inhibition halo. In plasticized PHB films, the bacterial inhibition halo against *S. aureus* was higher than that corresponding to unplasticized ones, comparing samples with the same concentration of copper (II) sulfate. This can be attributed to the reduction of intermolecular forces by the plasticizer, which improves the mobility of polymer chains, decreases the degree of crystallization, and increases the diffusion coefficient [38,39,40]. Regarding the effect of PHB films loaded with copper (II) sulfate on *P. aeuroginosa* growth, the minimum salt concentration that inhibited the bacterial development was 5% for unplasticized films (Figure 4b). In the case of PHB films plasticized with GTB, this minimum concentration was 2.5%, remarking again the effect of the plasticizer on the diffusion of the antimicrobial agent. There was a clear difference in the effect of copper (II) sulfate on the inhibition of Gram-positive and Gram-negative bacteria. Comparing the diameters of bacterial inhibition halos for both microorganisms at the same salt concentration, the halo of *S. aureus* (Gram-positive, Figure 4a) was higher than that corresponding to *P. aeruginosa* (Gram-negative, Figure 4b). This effect can be associated mainly with the differences in the structure of cell walls of both bacteria. Gram-positive bacteria have a thick layer of peptidoglycan and they do not have an outer lipid membrane. On the other hand, Gram-negative bacteria have a thin layer of peptidoglycan, sandwiched between an outer lipid membrane and the plasma membrane, resulting in a lipid bilayer structure. This structure acts as a molecular sieve that allows the diffusion of only relatively small molecules, generating some protection against several antibacterial agents [39,40].

### 3.4. Films’ Adhesiveness

Secondary dressing materials must meet one essential requirement: to be able to adhere to the skin. Thus, the PHB films’ adhesiveness was evaluated using different compression forces and the effect of copper salt on this property was studied. Figure 5a,b shows the system employed to evaluate the PHB adhesiveness to synthetic human skin. Tests showed that the films’ adhesiveness was not altered by copper (II) sulfate addition at a low level of compression force (Figure 5). At the higher assayed compression force (5 N) the films’ adhesive force was increased with the salt addition. This effect could be associated with the roughness generated by the presence of the antimicrobial agent. Ridges and valleys generated by the presence of copper (II) sulfate, which led to a greater surface area, could be responsible for the increase in the films’ adhesiveness.

### 3.5. Films’ Cytotoxicity

The evaluation of the toxicity of any wound dressing to healthy mammalian cells is critical for assessing its potential use in topical applications [41]. Cytotoxicity was evaluated through in vitro tests for unplasticized and plasticized samples, without and with 2.5% copper (II) sulfate, against L929 fibroblast cells. The cell viability of the different films is shown in Figure 6. Results indicated that all studied materials were not cytotoxic. According to ISO 10993 standard, cytotoxicity levels obtained for all materials were within acceptable ranges. Specifically, the standard considers noncytotoxic materials when the cell viability is in the range of 75 to 100%. Moreover, copper (II) sulfate did not affect in vitro film biocompatibility [42]. Cell viability exposed to PHB films was 100%. On the other hand, even when the cell viability of cells exposed to PHB-I-0Cu_GTB_ and PHB-I-2.5Cu_GTB_ was lower than that of the PHB films, values obtained were above the one where the material should be considered cytotoxic. Similar results in viability were reported [43] in PHB and could have an abundance of applications in the industry and medicinal sectors.

## 4. Conclusions

PHB films plasticized with glycerol tributyrate and functionalized with copper (II) sulfate resulted in noncytotoxic, adhesive, and antibacterial films, being attractive to be used as a secondary dressing for wound healing. All studied samples showed a good cell viability and low toxicity against L929 fibroblast cells. Plasticized PHB films with 0.5 and 2.5% copper (II) sulfate presented antimicrobial properties against *S. aureus* and *P. aeruginosa*, respectively. It was demonstrated that the presence of the plasticizer improved the diffusion of the copper (II) sulfate, giving enhanced antibacterial properties. On the other hand, PHB films showed a good adhesiveness to a synthetic skin sample by using a commercial adhesive, even with a very low compression force (1 N). Regarding mechanical properties, the presence of copper (II) sulfate decreased the mechanical strength and elongation at break of PHB films mainly due to the micrometric size of the crystals. An alternative to overcome this consequence could be to reduce the size of the salt particles prior to compounding. This work demonstrates that it is possible to formulate PHB-based materials through melt processing and thermocompression with specific properties for biomedical applications.

## Figures and Tables

**Figure 1 polymers-15-02907-f001:**
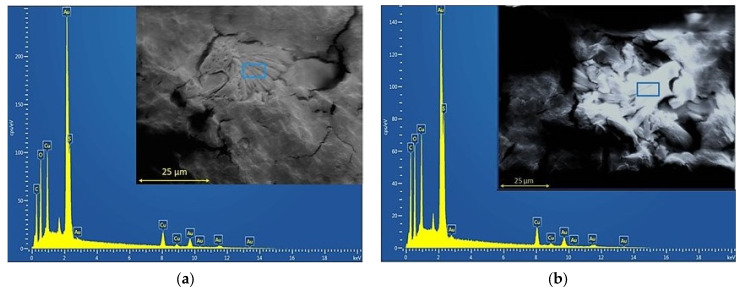
SEM micrographs and EDX spectra of PHB films without and with 20% GTB loaded with 2.5% copper (II) sulfate: (**a**) PHB-I-2.5Cu and (**b**) PHB-I-2.5Cu_GTB_.

**Figure 2 polymers-15-02907-f002:**
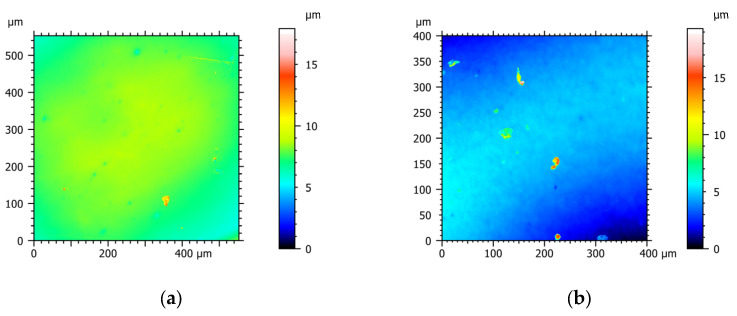
Two-dimensional AFM images of PHB without and with 2.5% copper (II) sulfate plasticized with 20% GTB: (**a**) PHB-I-0Cu_GTB_ and (**b**) PHB-I-2.5Cu_GTB_.

**Figure 3 polymers-15-02907-f003:**
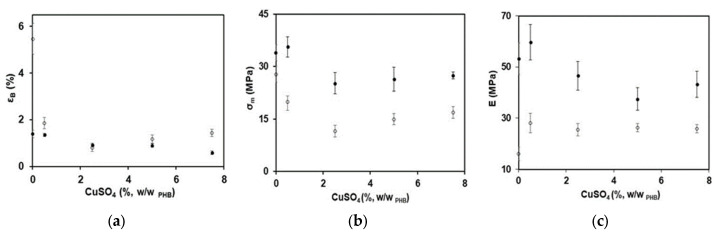
Mechanical properties of PHB films without and with 20% GTB loaded with different concentrations of copper sulfate: (**a**) elongation at break (ε_B_), (**b**) maximum tensile strength (σ_m_), and (**c**) Young’s modulus (E). Symbols: (⚫) PHB-I-Cu, (◯) PHB-I-Cu_GTB_. A statistical analysis of variance (ANOVA) was applied to compare the mean differences in mechanical properties by comparing the mean values with Fisher’s least significant difference test.

**Figure 4 polymers-15-02907-f004:**
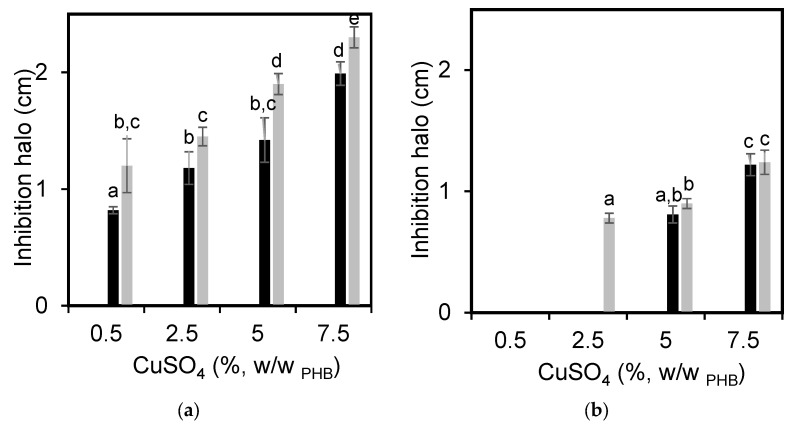
Inhibition halo of PHB films without and with 20% GTB loaded with different concentrations of copper (II) sulfate against: (**a**) *S. aureus* and (**b**) *P. aeruginosa*. Black bars represent PHB-I-Cu and gray bars represent PHB-I-Cu_GTB_. A statistical analysis of variance (ANOVA) was applied to compare the mean differences in inhibition halo by comparing the mean values with Fisher’s least significant difference test. Different letters indicate significant differences (*p* < 0.05).

**Figure 5 polymers-15-02907-f005:**
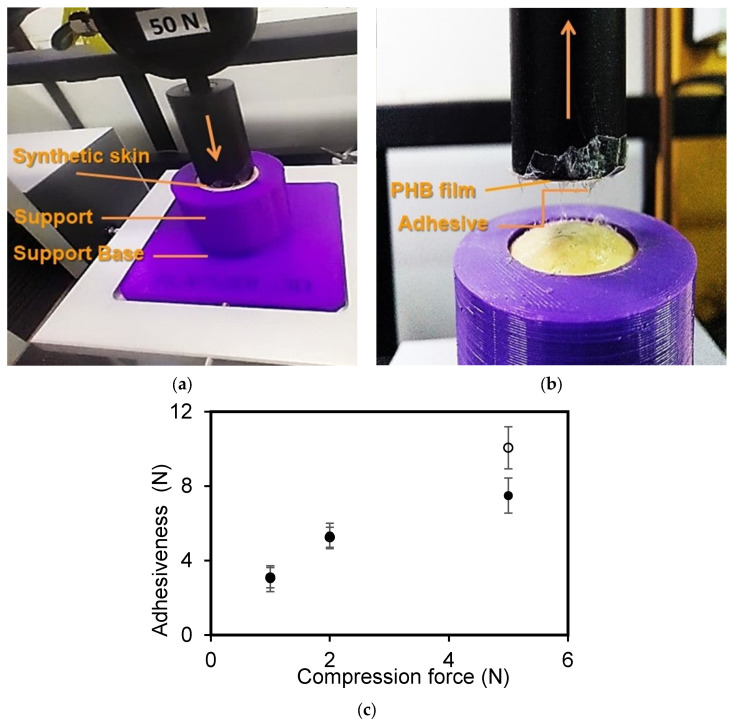
Adhesiveness tests of PHB films to synthetic human skin: (**a**) first stage: compression for 300 s; (**b**) second stage: test to register the maximum force to detach the film; and (**c**) adhesiveness of PHB films to synthetic human skin as a function of the compression force applied for 300 s. Symbols: (⚫) PHB-I-0Cu, (◯) PHB-I-2.5Cu_GTB_. A statistical analysis of variance (ANOVA) was applied to compare the mean differences in film adhesiveness by comparing the mean values with Fisher’s least significant difference test.

**Figure 6 polymers-15-02907-f006:**
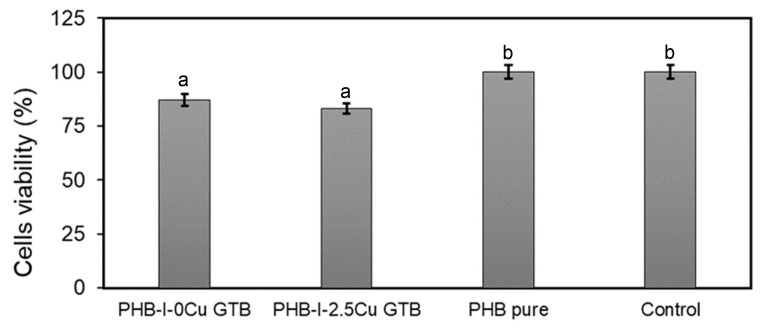
Cell viability of PHB films without and with 2.5% sulfate copper plasticized with 20% GTB. A statistical analysis of variance (ANOVA) was applied to compare the mean differences in cell viability by comparing the mean values with Fisher’s least significant difference test. Different letters indicate significant differences (*p* < 0.05).

**Table 1 polymers-15-02907-t001:** Formulations based on poly(3-hydroxybutyrate) (PHB), Irganox 1010 (I), glycerol tributyrate (GTB), and copper (II) sulfate pentahydrate (Cu).

Sample	Irganox 1010 (I) (%, w/w_PHB_)	Glycerol Tributyrate (GTB) (%, w/w_PHB_)	CuSO_4_·5H_2_O (Cu) (%, w/w_PHB_)
PHB-I-0Cu	0.3	-	-
PHB-I-0.5Cu	0.3	-	0.5
PHB-I-2.5Cu	0.3	-	2.5
PHB-I-5.0Cu	0.3	-	5.0
PHB-I-7.5Cu	0.3	-	7.5
PHB-I-0Cu_GTB_	0.3	20	-
PHB-I-0.5Cu_GTB_	0.3	20	0.5
PHB-I-2.5Cu_GTB_	0.3	20	2.5
PHB-I-5.0Cu_GTB_	0.3	20	5.0
PHB-I-7.5Cu_GTB_	0.3	20	7.5

## Data Availability

The data presented in this study are available on request from the corresponding author. The data are not publicly available due to privacy.

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
