# Peer review of "Films of Poly(Hydroxybutyrate) (PHB) and Copper with Antibacterial Activity"

_polymers, 2023, doi:10.3390/polym15132907_

Round 1
Reviewer 1 Report
The authors have done a study on the characterization of PHB films plasticized with glycerol tributyrate and functionalized with copper (II) sulfate. The study is interesting, but major improvements must be done before a possible acceptance.
- In the abstract, please specify against which species of bacteria and cell types were used in the study.
-In the introduction: Please correct the english for this sentence '' these materials are based on biopolymers which are capable of incorporate''
-In the introduction: please include more references for this statement; ''Innovation in antibacterial technology have led to the development of many modern 50 advanced wound dressings that incorporate antibacterial agents, such as silver, iodine, poly(hexamethylene biguanide) (PHMB), gentian violet, and methylene blue''
-In the introduction, try to avoid repetitions: e.g. bactericidal and bacteria are used in the same sentence ''Copper biocidal activity is a function of its concentration, acting as a bacterio static agent that inhibits bacterial growth at concentrations between 25 and 150 μM or 61 bactericidal that kills bacteria at higher concentrations''.
-In the introduction, please rephrase this sentence: In this work, it was obtained antimicrobial flexible films based on poly(3-hydroxy- 95 butyrate) (PHB), glycerol tributyrate as plasticizer, and cooper (II) sulfate as antimicrobial agent.
-In the paragraph 2.2, please improve the English for this sentence: ''Film samples were cryofracture...'' and please specify the country for the SEM. Moreover, change μm2 into μm2.
-In the paragraph 2.3, please specify the strain.
-In paragraph 2.4, please specify the bacterial product ID. (i.e. (ATCC® 25923). Change this 105 CFU mL-1 into 105 CFU ml-1. Please check the unit of measure after 200. Please specify for how long the films were disinfected in ethanol 70%. Why the authors didn't include a positive control (discs embedded with known antibacterial agents to compare the antimicrobial activity?
-In paragraph 2.5, please specify the origin (company?) from were the cells were bought. Check the past tense is used to describe the methods used. There is no need to include a lengthy description of how the MTT assay works. Please specify how you have grown the cells (medium). Please specify how you have made the materials sterile prior to testing. Please specify how many cells were used for each well and which passage. Please specify the initial stock concentration and the working concentration of MTT. Please specify the CO2% used in the study to grow cells. Please specify which instrument was used to read the absorbance.
-In paragraph 2.6 make sure the past tense is used to describe the methods. Please check the English for the paragraph (i.e. ''Then, a film was attached to a PLA support with a 197 similar diameter (17 mm) than the cap hole''). Why the films were were impregnated with a thin layer of a commercial adhesive for skin and hairpieces 196 MASTIC (PINTAFAN, Argentina)?
-In paragraph 3.2, please check the English is correct.
- In paragraph 3.3, please use italic to write the names of the bacteria.
-In paragraph 3.5, please specify which is the internal within a material is not cytotoxic ''values obtained were above the one from which the material should be considered cytotoxic''. How did you test the genotoxicity?
-Figure 5b, make sure everything is readable.
- Figure 7 must be improved. Please include the standard deviation and make the y axis larger (up to 120 % at least).
The Quality of English must be carefully checked by a fluent English speaker as the manuscript needs improvements.
Reviewer 2 Report
Dear Authors
In this work, it was obtained antimicrobial flexible films based on poly(3-hydroxybutyrate) (PHB), glycerol tributyrate as a plasticizer, and cooper (II) sulfate as an antimicrobial agent. Films properties relevant to secondary dressings of chronic wound healing, such as mechanical properties, adhesiveness, antimicrobial capacity, and cytotoxicity, were studied. Besides, film microstructure was also evaluated in order to understand the final properties. In all cases, the effect of plasticizer and copper (II) sulfate on film microstructure and final properties was carefully analyzed and discussed.
The following comments may help the authors clarify their presented work.
General comments
A) The authors dismissed Films properties relevant for secondary dressings of chronic wound healing, such as:
1- Water vapor permeability.
2- Oxygen permeability.
3- Hydrophobicity-hydrophilicity balance and subsequent absorbance of exudation.
B) The authors dismissed the study of copper sulfate particle size, which is relevant to the function of the developed wound dressing films.
Specific comments
Keywords: "Glycerol tributyrate plasticizer" must be added.
2.1. Materials and films obtaining
The authors select a fixed Glycerol tributyrate (GTB) concentration of 20% w/w of PHB. The authors need to justify why they chose this concentration. Why they did not study the effect of using different GTB concentrations?
2.4. Films antimicrobial capacity
The name of bacterial strains must be written in italics.
3.1. Films microstructure
"The corresponding micrographs and spectra are shown in Figures 2a and 2a, respectively." it should be corrected to "The corresponding micrographs and spectra are shown in Figures 2a and 2b, respectively."
Recommendations
A major revision is recommended.
A minor revision of the language is required.
Reviewer 3 Report
Suggested to modify the title, its wag and not reflecting the objective of work
Line no. 17: Please correct, properly use the abbreviation
Enlist the used bacteria with ATTCC/ or with detail from where collected in the methodology section or more suitably in material sections
Suggested to re-write the MTT cell viability assay method, by removing the unnecessary things. Diameter of disk placed ?
Suggested to discuss briefly why roughness increase after addition of copper in the film
Suggested to tabulate the mechanical property to make it more understandable for reader, also supplement the yield stress curve obtained from mechanical testing
The results of zone of inhibition are considered superficial without the pate image, suggested to supplement the image
Suggested to analysis the data properly using suitable statistic software
Suggested to re-plot the cell viability the why PHB pure and control not plated properly with statistic
Significant corrections required with proper presentation of the data with enrich discussion.
Minor editing of English language required
Round 2
Reviewer 1 Report
The authors have resolved most of the comments, but further minor changes are needed for improving the quality of the writing and the overall manuscript.
Please improve the English for this sentence in the introduction: ''In this work, it was obtained functional polymer films based on poly(3-hydroxy- butyrate) (PHB), glycerol tributyrate as plasticizer, and cooper (II) sulfate as antimicrobial 94 agent. Films properties relevant for secondary dressings of chronic wound healing such 95 as mechanical properties, adhesiveness, antimicrobial capacity, and cytotoxicity were studied''
Please, check again the English for this sentence (paragraph 2.2): ''Film samples were cryofracture, mounted onto bronze stubs, ''
Please, add the superscript for ''105'' CFU (paragraph 2.4).
Please, delete this sentence ''The tetrazolium reduction 158 assay was the first homogeneous cell viability assay developed for a 96-well format that 159 is suitable for high-throughput screening [26]''. It is not needed in this part.
Please, check the superscript for ''density of 0.01 x 10 6 cells/well.'' (paragraph 2.5)
Please correct the following statement: ''MTT stock was at 500 mg and was diluted to 5 mg/ml for the assays''. The stock should be given as a concentration, not as a mass. (paragraph 2.5)
Please, check your English for this sentence: ''The main objective of this section was to analyze the behavior of films that can be adhered to the skin. Wong et al. (1999) study the material adherence to a chicken skin using the same test. A commercial adhesive was used since PHB does not have adhesive capacity on the skin''
Please, improve the English for this sentence: (Paragraph 3.2) ''The incorporation of filler particles into polymeric matrices allows them to overcome certain materials limitations such as low rigidity and mechanical resistance in order to improve or expand their applications. ''
Please, include a description of the statistical analysis for each figure (in the legend and in the figure itself).
Please, improve the English for this sentence (in the conclusions): ''Despite all these promissory obtained results, the mechanical properties of films containing copper (II) sulfate should be improved by reducing the size of salt particles prior to compounding''
Please specify the company and characteristics of the human synthetic skin used in paragraph 2.6. Was the foam was placed underneath the human synthetic skin and the double sided tape or between the double sided tape and the PLA to provide a cushioning effect? Please, include an illustration on the setup used for the experiment as the provided photos are not sufficiently clear.
Please, for Figure 2, please add a low magnification image of the films and a higher magnification SEM image showing the presence of crystals, as the images are not very clear.
Please, change the authors contribution style: don't write Mayte M., but M. M. etc.
The English writing skills must be substantially improved before acceptance of the manuscript.
Author Response
The authors have resolved most of the comments, but further minor changes are needed for improving the quality of the writing and the overall manuscript.
1) Please improve the English for this sentence in the introduction: ''In this work, it was obtained functional polymer films based on poly(3-hydroxy-butyrate) (PHB), glycerol tributyrate as plasticizer, and cooper (II) sulfate as antimicrobial agent. Films properties relevant for secondary dressings of chronic wound healing such as mechanical properties, adhesiveness, antimicrobial capacity, and cytotoxicity were studied''
As it was suggested by the Editor, the aim of the work was rewritten.
2) Please, check again the English for this sentence (paragraph 2.2): ''Film samples were cryofracture, mounted onto bronze stubs, ''
The sentence was revised and rewritten.
3) Please, add the superscript for ''105'' CFU (paragraph 2.4).
It was added the superscript, changing 105 by 105.
4) Please, delete this sentence ''The tetrazolium reduction assay was the first homogeneous cell viability assay developed for a well format that is suitable for high-throughput screening [26]''. It is not needed in this part.
This sentence was deleted as well as the corresponding reference.
5) Please, check the superscript for ''density of 0.01 x 10 6 cells/well.'' (paragraph 2.5)
It was added the superscript, changing 0.01 x 10 6 by 0.01 x 106.
6) Please correct the following statement: ''MTT stock was at 500 mg and was diluted to 5 mg/ml for the assays''. The stock should be given as a concentration, not as a mass. (paragraph 2.5)
It was reported the concentration of the MMT stock.
7) Please, check your English for this sentence: ''The main objective of this section was to analyze the behavior of films that can be adhered to the skin.
We didn’t find this sentence in our revised manuscript
8) Wong et al. (1999) study the material adherence to a chicken skin using the same test. A commercial adhesive was used since PHB does not have adhesive capacity on the skin''
We already change this phrase in our version of round 1. Please check the new version of the manuscript.
9) Please, improve the English for this sentence: (Paragraph 3.2) ''The incorporation of filler particles into polymeric matrices allows them to overcome certain materials limitations such as low rigidity and mechanical resistance in order to improve or expand their applications.''
The sentence was revised and rewritten.
10) Please, include a description of the statistical analysis for each figure (in the legend and in the figure itself).
The description of the statistical analysis was included in Materials and Methods section and in the legend of each figure.
11) Please, improve the English for this sentence (in the conclusions): ''Despite all these promissory obtained results, the mechanical properties of films containing copper (II) sulfate should be improved by reducing the size of salt particles prior to compounding''
The sentence was revised and rewritten.
12) Please specify the company and characteristics of the human synthetic skin used in paragraph 2.6. Was the foam placed underneath the human synthetic skin and the double sided tape or between the double sided tape and the PLA to provide a cushioning effect? Please, include an illustration on the setup used for the experiment as the provided photos are not sufficiently clear.
The human synthetic skin used was a high-performance silicone rubber (Casiopea Beau-ty Store, CABA, Argentina). The company was added in the manuscript.
Double-sided tape was placed both between the PLA support and the foam and between the foam and the skin in order to prevent the slipping of materials during testing.
13) Please, for Figure 2, please add a low magnification image of the films and a higher magnification SEM image showing the presence of crystals, as the images are not very clear.
SEM images of now Figure 1 correspond to the zone where EDS was done. We do not have other figures with low and high magnification and the purpose of the image is to show the presence Cu salt in the film as it is shown in the EDS spectra.
14) Please, change the authors contribution style: don't write Mayte M., but M. M. etc."
Already done in the version round 2
Reviewer 2 Report
Dear Authors
The raised comments have been treated in a satisfactory way.
The revised version can be recommended for publication in its current form.
A minor revison of the language is recommended.
Author Response

(The authors gave the same response as above.)

Reviewer 3 Report
The authors have significantly improved the manuscript by incorporating the suggestions. I recommended editors that manuscript can be accepted in present from.
Thanks
The English Language is fine
Author Response

(The authors gave the same response as above.)

Round 3
Reviewer 1 Report
The authors have improved the manuscrpit but minor corrections are needed. There are still correction to be made in the test, for instance, in the intro, "are-capable "
Rephrase the sentence "new....devices", "all the experiments...by triplicate". "Cells without treatments....viable".
Fir the cytotoxicity studies, check the past tense is used to describe all the steps.
In figure there is no symbol indicating the stats.*,**,***. Include them. All the figures must indicate the stat differences between analysed groups.
Moderate English revision of the manuscript
Author Response
All the sentences marked by the reviewer were changed or rephrased.
The paragrph regarding cytotoxicity studies was corrected in the new version of the manuscript.
Stat differences between analysed groups were added in Figures 4 and 6.